# Mesenchymal Stem/Stromal Cells for Rheumatoid Arthritis Treatment: An Update on Clinical Applications

**DOI:** 10.3390/cells9081852

**Published:** 2020-08-07

**Authors:** Mercedes Lopez-Santalla, Raquel Fernandez-Perez, Marina I. Garin

**Affiliations:** 1Division of Hematopoietic Innovative Therapies, Centro de Investigaciones Energéticas, Medioambientales y Tecnológicas (CIEMAT) and Centro de Investigación Biomédica en Red de Enfermedades Raras (CIBER-ER), 28040 Madrid, Spain; Raquel.Fernandez@externos.ciemat.es; 2Advanced Therapy Unit, Instituto de Investigación Sanitaria Fundación Jiménez Díaz (IIS-FJD/UAM), 28040 Madrid, Spain

**Keywords:** rheumatoid arthritis, mesenchymal stem/stromal cells, clinical trials, cell therapy protocols

## Abstract

Rheumatoid arthritis (RA) is a chronic systemic autoimmune disease that affects the lining of the synovial joints leading to stiffness, pain, inflammation, loss of mobility, and erosion of joints. Its pathogenesis is related to aberrant immune responses against the synovium. Dysfunction of innate and adaptive immunity, including dysregulated cytokine networks and immune complex-mediated complement activation, are involved in the progression of RA. At present, drug treatments, including corticosteroids, antirheumatic drugs, and biological agents, are used in order to modulate the altered immune responses. Chronic use of these drugs may cause adverse effects to a significant number of RA patients. Additionally, some RA patients are resistant to these therapies. In recent years, mesenchymal stem/stromal cell (MSCs)-based therapies have been largely proposed as a novel and promising stem cell therapeutic approach in the treatment of RA. MSCs are multipotent progenitor cells that have immunomodulatory properties and can be obtained and expanded easily. Today, nearly one hundred studies in preclinical models of RA have shown promising trends for clinical application. Proof-of-concept clinical studies have demonstrated satisfactory safety profile of MSC therapy in RA patients. The present review discusses MSC-based therapy approaches with a focus on published clinical data, as well as on clinical trials, for treatment of RA that are currently underway.

## 1. Introduction

Rheumatoid arthritis (RA) is a chronic autoimmune disease that affects joints and connective tissues with associated vascular, metabolic, bone and psychological comorbidities. Chronic inflammation in RA is characterized by altered innate and adaptive immunity including immune responses against autoantigens, dysregulated cytokine networks, immune complex-mediated complement and osteoclast and chondrocyte activation [1,2].

The current therapies for RA can be classified into four categories: non-steroidal anti-inflammatory drugs (NSAIDs), corticosteroids, non-biologic disease-modifying anti-rheumatic drugs (DMARDs) and biologic DMARDs. Treatment of RA has undergone continuous changes over the past twenty years in which the development of drugs has been conducted in parallel to a deeper understanding of the pathogenesis of the disease [2,3]. NSAIDs are the most-frequently used treatments to mitigate the pain. In addition to this, different combinations of corticosteroids are used due to their potent anti-inflammatory effects. Non-biologic DMARDs, such as methotrexate, leflunomide, hydroxychloroquine, and sulfasalazine, protect joints by slowing down the inflammatory arthritis. More recently, biologic response modifiers that can be considered a type of DMARDs, aim to stop inflammation by depleting B cells or by blockade of inflammatory cytokine pathways, such as tumor necrosis factor α (TNFα) or interleukin (IL) 6 pathways. Fortunately, in the last few years earlier intervention with DMARDs and the availability of new medications have greatly improved the outcomes in a large proportion of patients affected by RA. However, despite advances in understanding the etiology of RA and the advent of novel biologic drugs, a substantial number of individuals with RA remain intolerant or resistant to these therapies [4,5].

Cell therapy with mesenchymal stem/stromal cells (MSCs) is a very attractive new approach to address unresolved treatment issues for patients with RA. The interest surrounding the field of MSCs was initially based on their capacity for self-renewal and regeneration of tissues and organs [6]. Subsequently, given their immunomodulatory properties, therapies with MSCs extended their therapeutic potential to chronic inflammatory processes.

Numerous immune responses and mechanisms of action have been described for the immunomodulatory effect of MSCs. MSCs are able to mediate potent immunoregulatory effects through induction of factors, such as indoleamine 2,3-dioxygenase, prostaglandin E2 [7], transforming growth factor β (TGF-β), and IL-10 [8], among others [9,10,11,12,13,14]. As a result, MSCs can dampen metabolic reprogramming of different types of immune cells, reduce the proliferation rate of actively dividing cells and inhibit the secretion of inflammatory cytokines. Furthermore, MSCs can promote monocyte differentiation into M2 macrophage [15,16,17,18]. MSCs can also inhibit monocyte differentiation into dendritic cells and skew them to a tolerogenic profile [19]. Moreover, effects of MSCs on suppression of T helper 1 (Th1) cells and Th17 proliferation [20] together with regulatory T cells (Tregs) expansion have also been documented in many inflammatory conditions [21].

At present, nearly a thousand clinical trials have used MSC-based therapies [22]. Among those around one hundred trials have been conducted for treatment of immune-mediated disorders. These trials started in 2004 [23] for immune-mediated diseases, such as graft vs. host disease [24], inflammatory bowel disease [25], multiple sclerosis [26], systemic lupus erythematosus [27,28], type I [29,30,31] and type II [32] diabetes, primary Sjögren syndrome [33], ankylosing spondylitis [34], and RA [35,36,37,38,39,40,41,42,43], among others. Currently the clinical application of MSC-based therapies is being extended to autoimmune hepatitis (NCT01661842 and NCT02997878), chronic autoimmune urticaria (NCT02824393) and refractory autoimmune thrombocytopenia [44] (NCT04014166) (Table 1).

In addition to their regenerative and immunomodulatory properties, MSCs have the advantage of being readily isolated and expanded in vitro. MSCs have a unique immunological profile that accounts for their immune privilege, which allows adoptive transfer in allogeneic recipients with minimal risk of rejection. This peculiar biological aspect facilitates the allogeneic use of MSCs in the clinic, thus presenting the possibility of cell bank establishment [45] from a rather limited number of MSC donors for widespread use in allogeneic unrelated recipients.

In this review we will summarize the extensive use of MSCs for the treatment of RA. First, the most relevant results achieved so far in preclinical studies that have paved the way for the development of clinical protocols. A systematic search of the published reports in peer-reviewed journals has identified nine completed studies related to the safety and efficacy of MSC-based therapy for RA. We also described the state-of–the-art in clinical trials that are currently underway. The information used in this review was extracted from ‘*www.ClinicalTrials.gov*’ and ‘*Pubmed*’ databases using the term “mesenchymal” and/or “stem” and/or “stromal” and/or “arthritis” and/or “arthritic” and/or “rheumatic” and/or “rheumatoid” for clinical trials registered up to June 2020. 

## 2. Preclinical Studies

MSC therapy has been established in nearly one hundred studies using experimental models of RA. MSC therapy can significantly reduce experimental arthritis induction and progression in the majority of experimental models tested. Among those, collagen-induced arthritis (CIA) in mice is the most widely used model to assess the efficacy of MSC therapy [46,47]. These promising results have paved the way for MSC therapy as a promising new treatment in human RA disease.

In general, based on the experimental data, the most suitable tissue source, major histocompatibility complex (MHC) context, route of delivery and dosage for MSC therapy in RA preclinical studies remain inconclusive. By far, the most prevalent source of MSCs is adult bone marrow (BM), followed by adipose tissue and, more recently, umbilical cord either from whole tissue or more specifically from umbilical cord blood (UC) [48,49,50]. Of late, MSCs have been isolated from alternative tissues [51], such as gingiva [52,53], amniotic fluid [54,55], placenta [56], synovial fluid or membrane [57], and nasal tissues (https://patentscope.wipo.int; patent number: WO2019022386, [58,59]). In all instances, the expanded MSCs matched with the minimal criteria defined by the International Society for Cellular Therapy (ISCT) [60] and have demonstrated similar efficacy in ameliorating experimental arthritis [47]. According to the ISCT criteria, the identification of MSCs relies on the combined expression of surface markers, such as CD73, CD90, CD105, CD71, CD44, CD106, and the lack of hematopoietic and endothelial markers (CD34, CD45, CD11b, CD14, and CD31). MSCs are isolated by adherence to plastic surfaces and are capable of differentiating into osteocytes, chondrocytes, and adipocytes in vitro.

Most of the RA preclinical studies have been conducted under xenogeneic condition followed by syngeneic MHC context [46,47]. The large majority of the studies claimed that MSC therapy has a benefit in the RA experimental models regardless of the MHC context. According to Liu L et al.’s meta-analysis [47], a better outcome was achieved using xenogeneic cells rather than allogeneic or syngeneic MSCs.

Intravenous (IV) and intraperitoneal (IP) delivery are the most widely used routes of administration in RA experimental models. Alternative routes of administration have been proposed, such as intranodal [61], intra-articular (IA) [62,63,64,65,66], peri-articular [67], intra-muscular [68], and subcutaneous [69], with good results. This supports the idea that MSC-based therapy is mainly mediated by systemic biological effects [70,71,72].

Numerous studies have shown that higher efficacy can be achieved when the infusion of MSCs is done during the early phases of the disease [62,73,74,75]. Although the most commonly used MSC dose has been 1 × 10^6^ MSCs per mouse, disagreement exists regarding the optimal cell dosage and a range of 0.1 × 10^6^ to 30 × 10^6^ MSCs per mouse in single or multiple infusions (up to 5 times) has been described [46,47].

In experimental models of RA, a decrease in antibodies against collagen together with a decline in inflammatory cytokine levels, such as interferon γ (IFN-γ), TNF-α, IL-4, IL-12, IL-1β, and IL-17, has been described [75,76,77]. Numerous studies have also demonstrated the key role played by several chemokines, such as macrophage inflammatory protein 1 (CCL4/MIP-1), monocyte chemoattractant protein 1 (CCL2/MCP-1), regulated upon activation, normal T cell expressed and secreted (CCL5/RANTES), and receptor activator of nuclear factor kappa-Β ligand (RANKL) [62,76,78]. In parallel, an increase in IL-10, IFN-γ-induced protein 10 (IP-10) and/or chemokine receptor 3-alternative (CXCR3) anti-inflammatory cytokine levels in serum and synovium have been reported [55,79]. Additionally, an increased frequency of Tregs and IL-10 and TGF-β-secreting T cells with a decrease frequency of IL-17 and IFN-γ-secreting T cells (Th17 and Th1 cells) in spleen and/or draining lymph nodes have also been observed in vivo [21,78,80]. Recently, extracellular vesicles (EVs) released from MSCs have emerged as key paracrine messengers that participate in the healing process influencing the local microenvironment with anti-inflammatory effects. The first evidence that MSC-derived EVs can exert an immunomodulatory effect in a preclinical model of RA was demonstrated by Cosenza and collaborators [81]. In this sense, the anti-inflammatory role of the MSC-derived EVs on T and B lymphocytes was observed regardless of the priming status of the MSCs used for their isolation. Whilst preclinical studies have demonstrated the therapeutic potential for MSC therapy in RA, optimization for their clinical use is an ongoing challenge and genetically engineered MSCs are being proposed to enhance their therapeutic potential [78,82,83,84].

In summary, MSC-based therapy can decrease the degree of arthritis inflammation down to 30% in the majority of experimental models of RA used, regardless of the tissue origin of the MSCs and the route of administration used. In addition to this, early infusion of MSCs during the induction phase of the disease and an average cell dose of 2–3 × 10^6^ MSCs per mouse using either single or multiple infusions of MSCs have shown efficient modulation of experimental RA.

## 3. Clinical Studies

In parallel to the promising results achieved in the preclinical studies, nine clinical trials have been completed and their results published. In addition to this, nine clinical trials are still active and as a consequence their clinical data are not publicly available yet. As a whole, these studies aim to evaluate the safety and the efficacy of MSC-based therapy in RA (Figure 1, Figure 2, Figure 3, Table 2 and Table 3).

### 3.1. Completed Studies

The first pilot RA clinical trial with MSC therapy was conducted in 2010 by the Stem Cell Research Center in Korea, and the results were published in 2011 [35]. In this study, 10 patients with different autoimmune diseases were enrolled. Four out of the 10 patients participating in the trial had RA. The authors developed a protocol to isolate and expand autologous adipose-derived MSCs (AD-MSCs) to provide sufficient number of cells that allow multiple infusions of AD-MSCs per patient (more than 10^9^ cells after 3 to 4 passages). MSC-based therapy was used as compassionate-use treatment for patients with RA for whom no other treatment option was available. Different amount of AD-MSCs (ranging from 2 × 10^8^ to 3.5 × 10^8^ cells/patient) were infused IV in a single, double, or quadruple dose. In two patients, in addition to the IV-infused AD-MSCs, a single IA infusion of AD-MSCs was administered locally (ranging from 1 × 10^8^ to 1.5 × 10^8^ cells per patient). All patients were monitored for 13 months. Clinical benefit after autologous AD-MSC infusions in RA patients measured by Visual Analogue Scale (VAS) and Korean Western Ontario McMaster (KWOMAC) scores was reported. Multiple AD-MSC infusions were associated with a higher efficacy of the MSC-based therapy. Importantly, the data showed that multiple infusions up to 8 × 10^8^ of AD-MSCs in a period of less than one month was safe as no adverse effects were observed. This study should be considered as the first proof-of-concept clinical study that has shown a satisfactory safety profile of autologous MSC therapy in RA patients with promising trends for clinical efficacy (Table 2).

Another pilot clinical trial with MSC therapy in RA patients was conducted at the Drum Tower Clinical Medical College of Nanjing Medical University in China, and the results were published in 2012 [36]. In this study, four RA patients with a long history of disease, up to 42 months, were enrolled. All RA patients were steroid dependent or had failed therapies with methotrexate, hydroxychloroquine, leflunomide, sulfasalazine, or at least one TNF-α inhibitor. In this case, allogeneic either from BM or UC tissue were used. RA patients were treated with a single IV infusion of 1 × 10^6^ MSCs/kg of body weight. The follow-up was extended up to 24 months. The RA patients continued with their immunosuppressant treatments during the study. No adverse events were observed following the infusion of the allogeneic MSCs in any of the RA patients. Although none of them achieved remission after MSC therapy, 3 patients had a European league against rheumatism (EULAR) moderate response for up to 7, 17, and 23 months, respectively. These patients also had a moderate response for up to 6 months measured by erythrocyte sedimentation rate (ESR), C reactive protein (CRP), disease activity score (DAS) 28, and VAS score. The fourth patient did not show any sign of EULAR response to MSC therapy. The authors hypothesized that the low MSC dose used in the study may explain the transient benefit observed in three out of four patients participating in the study.

The first randomized multicenter double-blind placebo-controlled dose-escalation phase Ib/IIa clinical trial with allogeneic AD-MSCs in RA was conducted by Tigenix in Spain in 2011 (NCT01663116). Results were presented in 2013 at the annual meeting of the American College of Rheumatology [85] and published in 2017 [39]. A higher number of refractory RA patients, up to 53, participated in the study. These RA patients also had a long history of disease (more than 13 years) and were resistant to at least two biologics with a DAS28-ESR > 3.2. Enrolled RA patients were grouped in three cohorts and received allogeneic expanded adipose-derived stem cells (eASCs, named C×611), with doses of 1, 2, or 4 × 10^6^ cells/kg of body weight, respectively, administered IV, at days 1, 8, and 15. Additionally, a placebo group receiving only Ringer’s lactate solution were included (randomization 3:1). All patients maintained small dose of DMARD, NSAID, and/or corticosteroid treatments, but no biologic treatments were administered. Patients were monitored for 6 months. 

Similarly to the previous proof-of-concept studies, the IV infusion of Cx611 was well-tolerated, with no evidence of dose-related toxicity. The most frequent treatment-related adverse effect observed in the trial was transient fever. In addition, good response based on the EULAR criteria, low DAS28-ESR, and CRP, in contrast to the placebo group for up to three months, was reported. As suggested by the authors, the very refractory profile of the RA patients included in the study may have hindered the beneficial effects of the MSC therapy. Due to the limited number of RA patients in each cohort no clear conclusion on the MSC dose-response could be drawn from the study. Additionally, no significant changes in circulating T cell populations, including FOXP3^+^CD4^+^ regulatory T cells, among cohorts were observed. Interestingly, 19% of RA patients generated MSC-specific anti-human leukocyte antigen I (HLA-I) antibodies without apparent clinical consequences. Anti-HLA-II antibodies were not found. This was the first study with allogeneic MSC-based therapy for RA that studied immunogenicity against the donor MSC cells [39].

In a study conducted at the Hospital of People’s Liberation Army Air Force in China, 172 patients with RA were recruited between 2010 and 2012 in a phase I/II clinical trial (NCT01547091). A first report was published in 2013 [37]. Additionally, 64 patients were monitored for three years and the long-term results were published in 2019 [42]. Similar to the previous studies, the RA patients enrolled in this trial had a long course of the disease, from 6 months to 35 years, with inadequate responses to conventional medication. Patients received a single dose of allogeneic UC-MSCs isolated from the entire UC tissue (4 × 10^7^ cells/patient) IV infused. All patients also maintained low-dose DMARD treatments (leflunomide, hydroxychloroquine sulfate or methotrexate). MSC-treated RA patients were compared with RA patients treated only with DMARDs. In this study, no serious adverse effects were reported. Only 4% of the patients showed mild adverse effects, such as flu-like symptoms, upon infusion of the MSCs, which disappeared within hours. Interestingly, a significant remission of disease was observed according to the American College of Rheumatology improvement criteria (ACR), the DAS-28, ESR, and the Health Assessment Questionnaire (HAQ). This was accompanied by decreased levels of CRP, rheumatoid factor (RF), and anti-cyclic citrullinated peptide (anti-CCP) antibodies, as well as TNF-α and IL-6 cytokines in sera. The percentage of regulatory CD4^+^ T cells in peripheral blood was increased. No such benefits were observed in the control group of RA patients treated with DMARDS only. The beneficial effects were observed both in the short-term (up to 8 months) [37], as well as in the longer term, for up to 3 years [42]. This study demonstrated for the first time the long-term beneficial effects of MSC-based therapy in combination with low dose of DMARDs for patients with RA. Although a large number of patients with RA were enrolled in this study, it would be desirable to conduct a multiple-center clinical trial in order to confirm the promising outcome achieved in the study.

Another Chinese study was conducted at Daping Hospital, where 53 refractory RA patients were recruited from 2016 to 2017. These patients had failed to respond to DMARDs, NSAIDs, corticosteroids and biologics or could not tolerate their serious side effects. A control group of 53 patients treated with saline solution was included in the study. This phase I clinical trial (ChiCTR-ONC-16008770 in the Chinese registry; ‘*www.Chictr.org*’) aimed to determine the clinical efficacy of MSC therapy in RA patients and to identify a possible biomarker for predicting the beneficial effects of MSC therapy. Patients were treated with a single IV dose of 1 × 10^6^ allogeneic UC-MSCs/kg of body weight isolated from the UC tissue following blood vessel removal. The follow-up was for up to 12 months. The results, which were published in 2018 [40], showed no serious acute adverse events. Only three patients had chills or fever, which is a common event also observed in previous clinical trials. This study confirmed the clinical safety and efficacy of MSC therapy for the treatment of RA patients. Similarly, the clinical efficacy of MSCs varied greatly. Thus, 54% of the RA patients treated with MSCs achieved a good or moderate response, whereas 46% of RA patients had no clinical response during 12 months of follow-up as compared with the control group. The biological effect of the MSC-based therapy was measured by CRP, ESR, and the HAQ and DAS28 scores. In contrast to the previous study (NCT01547091), the effects of MSCs were transient. Around 8% of RA patients experienced a relapse in the disease by 24 weeks with good or moderate response. The authors suggested that an additional MSC infusion 6 months later may be required in order to achieve a sustained effect upon infusion of the MSCs. The RA patients who had a positive response showed increased levels of albumin, hemoglobin, and IL-10 in sera. This was accompanied by a decrease in platelet, RF, anti-CCP, IL-6, and TNF-α cytokines. Additionally, an increased percentage of regulatory CD4^+^ T cells and a concomitant decrease in Th17 were observed in the responder RA patients. These results were similar to what was observed in the previous study conducted in China [37,42]. No significant changes in IL-1β, IL-2R, and IL-8 levels were reported. High serum IFN-γ levels were observed before and 4 weeks after the infusion of the MSCs in the responder RA patients in comparison to the low levels observed in the non-responder RA patients. The authors associated the high serum IFN-γ levels positively with the reduced DAS28 value in the responder RA population, claiming that serum IFN-γ levels could be used as a biomarker to predict a clinical benefit for the patients. These data are in line with the preclinical studies suggesting that MSCs exerted their immunosuppressive effects when the MSCs encounter an inflammatory environment within the host. These encouraging results should be further confirmed in a multicenter study (Table 2). Based on these observations, the same investigators conducted a preclinical study where CIA mice were treated with INF-γR^−/−^ MSCs and their in vivo efficacy at modulating progression of experimental RA was compared to wild-type MSCs. Cell therapy treatment of arthritic mice with INF-γR^−/−^ MSCs had no therapeutic effects on CIA progression suggesting that the therapeutic efficacy of MSCs on RA is highly dependent on the IFN-γ levels and that blockade of IFN-γ signaling on MSCs successfully undermined their immunomodulatory effects. These observations further agree with the widely accepted concept that MSC immunosuppressive properties are not constitutive. Instead, their immune regulation depends on a process of “licensing”, which needs to be acquired in an inflammatory microenvironment. In 2017, Xu and collaborators registered a second clinical trial (ChiCTR-INR-17012462; Table 2), where 63 refractory patients with RA were treated with UC-derived MSCs combined with recombinant IFN-γ. The outcome of the study revealed that combination of MSCs plus IFN-γ greatly improved the clinical efficacy of MSC-based therapy in RA patients from 53.3% to 93.3%. No unexpected safety issues were observed 1-year follow-up in any of the patients participating in the study [86].

From 2016 to 2017, 9 RA patients, who were refractory to standard therapies, were recruited by the Imam Reza Hospital in Iran for a phase I clinical trial (NCT03333681). The enrolled RA patients were IV-infused with a single dose of autologous BM-MSCs (1 to 2 × 10^6^ cells/kg of body weight). All patients continued to receive conventional therapy (different combinations of sulfasalazine, prednisolone, methotrexate, and/or hydroxychloroquine). RA patients were monitored for up to 12 months. No complications or adverse events were observed following the infusion of the BM-MSCs in any of the RA patients. A significant decrease in DAS28-ESR, VAS, and ESR up to 12 months was reported [43]. In contrast to previous clinical trials using MSC-based therapy for RA, no statistically significant differences in serum levels of CRP, anti-CCP, IL-17, and IFN-γ cytokine levels were observed [43,87]. A significant increase in the percentage of regulatory CD4^+^ T cells one month after MSC infusion was observed. The increase of regulatory CD4^+^ T cells measured by FOXP3 mRNA levels was maintained in parallel to an increase in T-bet and GATA3 transcription factor mRNA levels and IL-10 and TGF-β cytokine levels in the sera 12 months after the MSC infusion [87]. The authors suggested that the decrease in percentage of regulatory CD4^+^ T cells in the periphery following MSC infusion suggests that multiple infusions of MSCs or higher dose of MSCs may be required to sustain expansion of in vivo regulatory CD4^+^ T cells. Moreover, a year post-infusion of the BM-MSCs, a significant decrease in the percentage of Th17 cells was observed. These results were in line with the results found by Wang L et al. [37,42] and Yan Y et al. (ChiCTR-ONC-16008770) [40]. They also observed an inverse correlation between the IL-4 cytokine levels in the sera and the DAS28 at 6 months follow-up [87]. Furthermore, in this clinical trial the authors evaluated for the first time the response of B cells, B-cell activating factor (BAFF), a proliferation-inducing ligand (APRIL), and their receptors on the surface of B cells, such as B cell activation factor receptor (BR3), transmembrane activator and CAML interactor protein (TACI), and B-cell maturation antigen (BCMA). A decrease in the plasma levels of CD19^+^ B cells with a decreased expression of BR3, TACI, and BCMA receptors 12 months after the MSC infusion was observed. Additionally, a decrease in the BAFF plasma levels and APRIL plasma levels after were also measured months after the infusion of the MSCs. These results indicate that MSC therapy decreases the levels and proliferation of B cells for at least one year following the MSC treatment in the RA patients [88]. Furthermore, the plasma levels of CXCL8, CXCL12, and CXCL13 were significantly decreased 6 months after MSCs transplantation, suggesting that the interaction of MSCs with CXCL8-producing cells could be one of the possible mechanisms causing the reduced levels of CXCL8 in the joints and, subsequently, in the plasma of RA patients. CXCL8 reduction returned to pre-treatment levels after 12 months, thus suggesting that an increase dose of MSCs or multiple injections of MSCs may result in a sustained anti-inflammatory effects of MSCs [89].

Kang Stem Biotech, in Korea, has funded two clinical trials (phase I NCT02221258, named CURE-IV; and phase II NCT03618784, named FURESTEM-RA; Table 2 and Table 3, respectively). These trials were conducted at Seoul Metropolitan Government-Seoul National University Boramae Medical Center. In contrast to the clinical trials mentioned above, nine RA patients that had not previously been treated with any biologic compound were enrolled. RA patients were treated IV with allogeneic UC-MSCs isolated from UC blood. In the phase I clinical trial, the MSC doses were 2.5 × 10^7^, 5 × 10^7^ or 1 × 10^8^ UC-MSCs per subject [41]. Clinical and safety parameters were monitored for one month following the infusion of the MSCs. No serious adverse events or major abnormalities in serum chemical or hematologic profiles were observed. A decline in the DAS28-ESR, HAQ, and VAS score was reported. Reduced serum levels of ESR and CRP were measured. IL-1β, IL-6, IL-8, and TNF-α levels were reduced at 24 h in the group infused with the higher dose of MSCs (1 × 10^8^ MSCs per subject). A statistically significant increase in serum levels of IL-10 was found at 24 h post-infusion of 5 × 10^7^ MSCs. Reduced serum levels of IL-6 and TNF-α and increase in IL-10 serum levels were also observed in previous RA clinical trials with MSC therapy [37,40,42]. This study, although interesting, had some limitations, such as the relatively small number of patients enrolled and the rather limited 4-week follow-up period following the infusion of the MSCs. The same group is currently conducting a 5-year observational study on these patients. Interestingly, a multicenter randomized double-blind parallel placebo-controlled phase I/IIa RA clinical trial with MSC therapy (NCT03618784) has been initiated in the same center (Table 3).

In 2013, Mesoblast Ltd. launched a multicenter randomized double-blind placebo-controlled sequential dose-escalation phase II clinical trial in the USA (NCT01851070). In this study, the so-called allogeneic multipotent progenitor cells (MPCs), identified by the surface expression of STRO-1^+^ or STRO-3^+^ markers from BM-derived cells, were used. These MPCs had increased clonogenic, developmental, and proliferative capacity compared with unfractionated MSCs [90]. In this study, 1 × 10^6^ or 2 × 10^6^ MPCs/kg of body weight were IV-infused in 48 patients with active RA who also received methotrexate or alternative DMARDs for at least 6 months prior to screening and who had had an incomplete response to at least one TNF-α inhibitor. A placebo group of RA patients was included in the study. The follow-up period was extended for up to 52 weeks. The safety and efficacy outcomes of the clinical trial over the 12-week primary evaluation period were presented at the EULAR annual European congress of rheumatology in June 2017 [38]. A single IV infusion of MPCs was well-tolerated. The authors also reported improvement in clinical symptoms, physical function, and disease activity measured by ACR, patient global assessment (PGA), and HAQ. The highest MPC dose used (2 × 10^6^ MPCs/kg of body weight) showed the earliest and most sustained treatment benefit. No other data have been published so far. As suggested by the authors, while the efficacy results were encouraging, further assessment including dose optimization will be required. The current trial is the only RA clinical trial that has showed results using MPCs as a therapeutic option in biologic-refractory RA patients. Currently, a clinical trial has been initiated using MPC-based therapy for RA patients in the USA (NCT03186417) (Table 3).

Based on the completed and published RA clinical trials using MSC-based therapy, promising results in terms of safety and efficacy have been observed. Strikingly, these encouraging results have been obtained in refractory patients with long histories of RA. No adverse effects and only a moderate response to the MSC therapy have been observed in all the RA clinical trials conducted so far. Moreover, efficacy observed during 3 years of follow-up [42] pointed out the long-term effect of MSCs in RA patients in agreement with what has been observed in other immune-mediated diseases [18,91].

Today, there is no optimal protocol for MSC therapy in RA patients, although, in terms of MHC context, most of them (78%) used allogeneic MSCs due to the difficulty in isolating and expanding a large enough number of autologous MSCs (Figure 1B). The use of allogeneic MSCs is based on the low immunogenicity of MSCs that implies the possibility of greater availability via cell banks establishment. This is in contrast to the general protocol in clinical trials using MSC therapy for treatment of other types of pathologies in which autologous and allogeneic contexts are used in similar proportion [22]. MSCs from BM, AD, and UC have been used (30%, 20%, and 50%, respectively, Figure 1C) with similar results in terms of safety and efficacy. In clinical trials using MSC-based therapy for other types of inflammatory diseases, BM is the main tissue source followed by UC [22]. More than half of the clinical trials for treatment of RA have infused less than 10 × 10^6^ MSC/kg of body weight in a single infusion (Figure 1D,E). Although, based on dose-escalation RA clinical trials, both the cell dose and the use of multiple infusions of MSCs did not seem to correlate to their beneficial effect. Therefore, several groups have pointed out that cell doses above 1 × 10^6^ cells/kg of body weight could be better in terms of short and long-term efficacy. This is in line with Kabat et al.′s report [22], in which lower efficacy was observed with MSC doses lower than 70 million cells/patient (~1 × 10^6^ MSC/kg of body weight). In the highest MSC dose tested so far (8 × 10^8^ MSCs/patient), no adverse effects have been reported, thus confirming that a wide range of MSC dosage can be tolerated.

Despite the encouraging results obtained so far, most of the clinical trials conducted have enrolled low number of patients. A placebo group is also lacking in some of the studies. The large majority of the patients enrolled in these trials have a long history of RA refractory to conventional treatment. This is in sharp contrast to the pre-clinical, as well as clinical studies for other inflammatory pathologies where treatment during the early phases of the disease seems more effective.

### 3.2. Active Clinical Trials

Based on the study conducted by Wang et al. [37,42], in 2013, the Stem Cell Institute in Panama has launched a phase I/II clinical trial using allogeneic UC-MSCs from UC tissue for treatment of 20 DMARD-resistant RA patients (NCT01985464; Table 3). The authors of the study aimed to define treatment-associated adverse events and biological efficacy measure by CRP, ESR, anti-CCP, RF, HAQ, DAS28, and EULAR response criteria and immunological parameters one-year post-infusion. This clinical trial is underway and the estimated study completion date is June 2020.

In 2016, an interventional clinical trial sponsored by Stem Cells Arabia (Amman, Jordan) was registered (NCT03067870). The investigators aimed to define the safety and efficacy of autologous BM-MSCs administered IV, as well as IA, in the joints of patients with RA. The authors aimed to enroll 100 RA patients. The patients participating in the study will be monitored for a month using VAS scoring in order to evaluate the systemic efficacy of MSC therapy. The regenerative and repair potential of BM-MSCs in joints will be monitored for up to 6 months by analyzing the physical activity of the patient using the Western Ontario and McMaster Universities Osteoarthritis Index (WOMAC) score and magnetic resonance imaging (MRI). The estimated study completion date is February 2022.

In January 2019, a multicenter randomized controlled clinical study based at the Xijing Hospital in China was registered and is currently recruiting patients (NCT03798028). This study aims to evaluate the safety and therapeutic effects of a single-dose of human allogeneic UC-MSCs isolated from UC blood on adult patients with moderate or severe RA who suffer from anemia or/and interstitial pulmonary disease. Half of the participants will receive a single dose of 1 × 10^6^ cells/kg of body weight of allogeneic UC-MSCs together with their present medication, while the other half will receive a placebo combined with their present medication. This is a unique clinical trial that is enrolling a specific group of 250 DMARD-resistant RA patients with an associated disease. The safety and therapeutic effects of the MSC-based therapy will be monitored for 24 weeks. The estimated study completion date is June 2020.

A proof of concept phase I clinical trial conducted at the MetroHealth Medical Center Cleveland, OH, USA, was registered in 2017 (NCT03186417). This is the first multicenter, double-blind, placebo controlled interventional clinical trial that is recruiting RA patients with new onset of the disease (diagnosis ≤1 year and symptoms for ≤2 years). A total of 20 RA patients will be enrolled. RA patients that previously took DMARDs other than non-steroidal, prednisone, hydroxychloroquine, and methotrexate included leflunomide and biological treatments will be excluded from the study. This clinical trial will evaluate the safety and efficacy of allogeneic BM-MSCs (2, 4, or 6 × 10^6^ cells/kg of body weight) over 24 months. This clinical trial aims to infuse the highest allogeneic BM-MSC dose tested so far in a trial (Table 2 andTable 3). Safety and efficacy will be monitored by PROMIS computer-adaptive testing (CAT) instruments (pain interference, physical function, sleep disturbance, and fatigue) and Routine Assessment of Patient Index Data 3 (RAPID3) questionnaires and DAS28-CRP. Recruitment is currently open and final data collection date for primary outcome is estimated by December 2020.

In 2018, Kang Stem Biotech Co., Ltd. began a multicenter randomized double-blind parallel placebo-controlled phase I/IIa RA clinical trial (NCT03618784) in South Korea with IV administration of allogeneic UC-MSCs isolated from UC blood (called FURESTEM-RA Inj) in 33 RA patients (diagnosed at least 12 weeks before with DAS28-ESR > 3.2) from the Seoul Metropolitan Government-Seoul National University Boramae Medical Center. The RA clinical trial characteristics are based on a phase I clinical trial conducted by the same group in 2014 (NCT02221258). In this study, the authors included RA patients that were refractory or intolerant to DMARDs and biologic DMARDs. Two doses of allogeneic UC-MSCs (5.0 × 10^7^ or 10 × 10^7^ cells/patient) infused three times IV with 4 weeks interval between MSC infusions will be tested. Patients will be monitored for 16 weeks. Safety and efficacy of MSC-based therapy will be analyzed by ACR, EULAR, DAS28-ESR, Korean Health assessment questionnaire (KHAQ), clinical disease activity index (CDAI), and VAS scores. Additionally, a panel of cytokine levels will be monitored in sera. This clinical trial is now active and recruiting patients. The estimated study completion date is April 2021.

Since 2018, Hope Biosciences in Texas, USA, has been conducting a phase I/II clinical trial (NCT03691909) using a single dose of autologous AD-MSCs, enrolling RA patients at diagnosis or RA patients who are on a stable dose of therapy regimen for more than 4 weeks prior to screening. Together with the NCT04170426 trial registered by Celltex in 2020 that is enrolling DMARD-resistant RA patients, this is the second active RA clinical trial using autologous AD-MSCs. The study’s purpose is to evaluate the safety and efficacy of MSC therapy up to 12 months post-infusion. The parameters that will be used to monitor the efficacy of the MSC-based therapy are CRP, ESR, and 68 joint assessments, together with TNF-α and IL-6 serum levels. The recruitment of patients has been completed, and the follow-up is in progress. The estimated study completion date is June 2020.

Baylx Inc. of Irvine, California, USA, has recently launched a phase I, randomized (3:1), placebo-controlled, double-blind, single-dose clinical trial (NCT03828344) based on the NCT01547091 clinical trial [37,42]. The aim of the study is to define the safety and biological effects of a single dose of fresh allogeneic UC-MSCs named BX-U001 isolated from entire UC tissue (0.75 or 1.5 × 10^6^ cells/kg of body weight), given by IV infusion to 16 refractory RA patients. Additionally, RA patients will be monitored for 12 months. During the study, RA patients will continue to be treated with their previous conventional DMARDs and NSAIDs, other than biologics. RA patient efficacy will be determined by HAQ, ESR, CRP, DAS28-CRP, simplex disease activity index (SDAI), and ACR criteria. In addition, RF and anti-CCP serum levels will be measured. This RA clinical trial is currently opened but recruiting has not begun yet. The estimated study completion date is September 2022.

Very recently, Celltex Therapeutics Corporation in Houston, USA, registered a dose escalation randomized placebo controlled and double-blind phase I/IIa clinical trial (NCT04170426). Up to 54 active DMARD-resistant RA patients will receive one or three doses of 2.0–2.86 × 10^6^/kg of body weight of autologous AD-MSCs IV on days 1, 4, and 7. The follow-up period will be 12 months. The efficacy of the MSC-based therapy will be evaluated by ACR, RAPID3, DAS28-CRP, and an array of peripheral blood inflammatory panel of cytokines. The estimated study completion date is December 2023.

In summary, 8 RA clinical trials using MSC-based therapy are ongoing with a great heterogeneity in terms of MHC context, tissue source, and cell dose used (Figure 2 and Table 3), meaning that at present no consensus on the optimal protocol exists. Although most of these trials are being conducted in an allogeneic context (62.5%, Figure 2B), the use of autologous cells has increased with respect to completed RA clinical trials. The main tissue source used to isolate and expand MSCs is umbilical cord tissue, which is used in nearly 50% of active studies (Figure 2C), although, in the past, bone marrow was the chosen tissue to generate MSCs for clinical use (Figure 1B,C and Figure 2B,C). Interestingly, RA patients in the early phases of the disease prior to biological treatments are now being recruited (Figure 2A). This change in the inclusion criteria may increase the efficacy of MSC therapy since a large majority of RA patients treated with MSCs were refractory to conventional treatments and had a long history of disease. In the active RA clinical trials with MSC therapy, control groups are now being included, which increases the value of the results obtained. In general, the follow-up proposed is up to 12 months. Additionally, no changes in the efficacy criteria are included since the parameters that have been used until now are widely recognized and accepted by the official RA worldwide associations (ACR and EULAR).

## 4. Conclusions

Clinical trial registrations in RA patients with MSC therapy have increased linearly from 2011 to today (Figure 3). In general, new registrations of clinical trials with MSC-based therapy have reached a plateau since 2018 [22]. No toxicity and adverse effects have been found in any of the RA clinical trials conducted. No sufficient data on efficacy have been obtained from the completed studies, most likely due to the fact that the large majority of the RA patients enrolled in these studies were refractory to conventional RA treatments with a long history of the disease. Thanks to the good safety profile of MSC-based therapy for RA at present, there are eight clinical trials using MSC-based therapy registered and active in ‘clinicaltrials.gov’ where MSC treatment at early stages of the disease are being explored. For better comparisons of results among RA clinical trials with MSC-based therapy, an improvement in the standardization of MSC treatment protocols in terms of manufacturing protocols, sources of MSCs, MHC contexts, routes of delivery, cell dosing, and systematic analysis of the results will be needed. Additionally, the identification of RA patients most likely to respond to MSC treatment will clearly benefit the clinical application of MSC-based therapies for RA.

## Figures and Tables

**Figure 1 cells-09-01852-f001:**
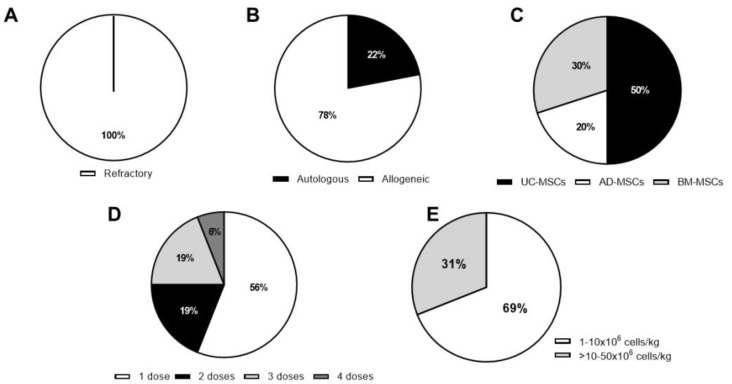
Characteristics of the completed rheumatoid arthritis (RA) clinical trials. (**A**) RA disease status of patients; (**B**) major histocompatibility complex (MHC) context (**C**) MSC tissue source. Umbilical cord (UC-MSCs), adipose tissue (AD-MSCs), and bone marrow (BM-MSCs); (**D**) number of doses and (**E**) MSC dose expressed as number of cells/kg of body weight (1–10 × 10^6^ or >10–50 × 10^6^). Data are represented as percentage of the total number of studies.

**Figure 2 cells-09-01852-f002:**
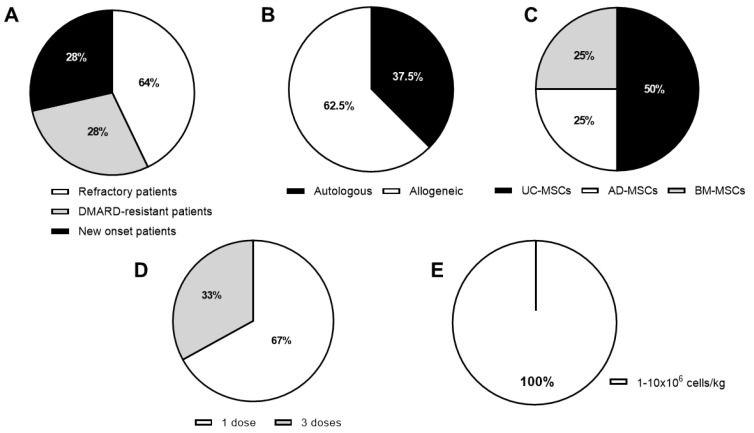
Characteristics of the open RA clinical trials. (**A**) RA disease status of patients; (**B**) MHC context (**C**) MSC tissue source. Umbilical cord (UC-MSCs), adipose tissue (AD-MSCs) and bone marrow (BM-MSCs); (**D**) number of MSCs infusions and (**E**) MSC dose expressed as number of cells/kg of body weight (1–10 × 10^6^). Data are represented as percentage of the total number of studies.

**Figure 3 cells-09-01852-f003:**
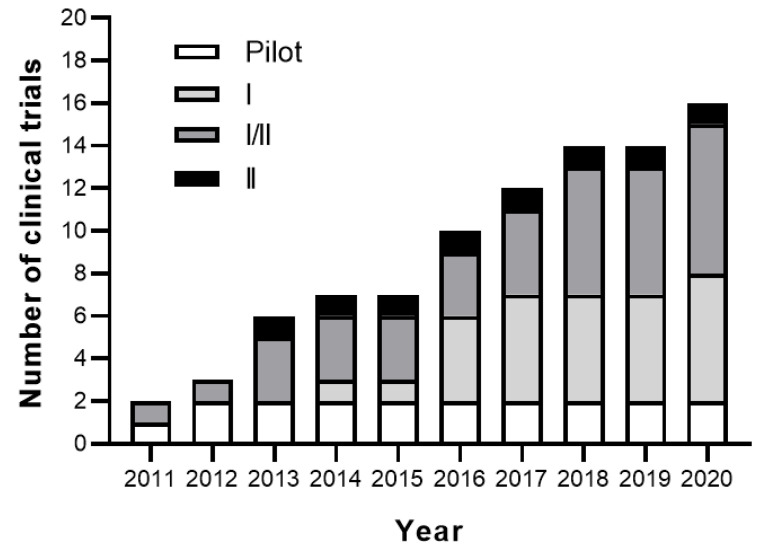
Cumulative number and phase of registered clinical trials with MSC-based therapy in RA patients according to the year of registration in ‘*www.ClinicalTrials.gov*’ and ‘*Pubmed*’ databases.

**Table 1 cells-09-01852-t001:** Immune-mediated disorder clinical trials using mesenchymal stem/stromal cell (MSC) therapy.

Immune-Mediated Disorders	Number of Clinical Trials	Year of First Clinical Trial	References
Graft vs. host disease	49	2004	[24]
Inflammatory bowel disease	23	2006	[25]
Multiple sclerosis	29	2006	[26]
Systemic lupus erythematosus	10	2007	[27,28]
Type I diabetes	26	2008	[29,30,31]
Primary Sjögren syndrome	1	2009	[33]
Type II diabetes	13	2010	[32]
Autoimmune hepatitis	2	2011	NCT01661842 and NCT02997878
Ankylosing spondylitis	2	2011	[34]
Chronic urticaria	1	2017	NCT02824393
Refractory autoimmune thrombocytopenia	1	2019	NCT04014166

**Table 2 cells-09-01852-t002:** Summary of completed rheumatoid arthritis clinical trials with MSC therapy.

Clinical Trial Identifier	Clinical Phase	Source	Registration Year	Country	RA Patients	MHC Context; Route of Administration	Cells/kg of Body Weight; Number of Doses and Route of Administration	Number of RA Patients Enrolled	Follow-Up (Months)	Publication (Year)	Included Control Group
Unknown	Pilot	AD	2011	Korea	Refractory	Autologous; IV and IA	3 × 10^8^/patient; 2 doses 2 × 10^8^/patient; 4 doses 2 × 10^8^/patient; 1 dose IV + 1 × 10^8^/patient; 1 dose IA 3.5 × 10^8^/patient, 1 dose IV +1.5 × 10^8^/patient; 1 dose IA	4 out of the 10 patients enrolled	12	[35]	No
Unknown	Pilot	BM/UC	2012	China	Refractory	Allogeneic; IV	1 × 10^6^ 1 dose	4	19	[36]	No
NCT01663116	Ib/IIa	AD	2011	Spain	Refractory	Allogeneic; IV	1, 2 or 4 × 10^6^; 3 doses, weekly	53	6	[39]	Yes
NCT01547091	I/II	UC	2013	China	Refractory	Allogeneic; IV	4 × 10^7^/patient; 1 dose	172	36	[37,42]	Yes
ChiCTR-ONC-16008770	I	UC	2016	China	Refractory	Allogeneic; IV	1 × 10^6^; 1 dose	53	12	[40]	Yes
NCT03333681	I	BM	2016	Iran	Refractory	Autologous; IV	1 to 2 × 10^6^; 1 dose	15	12	[43]	No
NCT02221258	I	UC	2014	Korea	Refractory	Allogeneic; IV	2.5, 5 or 10 × 10^7^/patient; 1 dose	9	1	[41]	No
NCT01851070	II	MPCs	2013	USA	Refractory	Allogeneic; IV	1 or 2 × 10^6^; 1 dose	48	3	[38]	Yes
ChiCTR-INR-17012462	I/II	UC	2017	China	Refractory	Allogeneic; IV	1 × 10^6^; 1 dose	63	3	[86]	No

AD, adipose tissue; BM, bone marrow; IA, intra-articular; UC, umbilical cord; IV, intravenous; MPCs, multipotent progenitor cells.

**Table 3 cells-09-01852-t003:** Summary of active RA clinical trials with MSC therapy.

Clinical Trial Identifier	Clinical Phase	Source	Registration Year	Country	Status	RA Patients	MHC Context; Route of Administration	Cells/kg of Body Weight; Number of Doses	Estimated Number of RA Patients	Follow-Up (Months)	Included Control Group	Estimated Completion Date
NCT01985464	I/II	UC	2013	Panama	Active, not recruiting	DMARD-resistant	Allogeneic; IV	Unknown	20	12	No	June 2020
NCT03067870	I	BM	2016	Jordan	Active, not recruiting	Unknown	Autologous, IV and IA	Unknown	100	6	No	February 2022
NCT03798028	N/A	UC	2017	China,	Recruiting	Anemia or pulmonary disease associated	Allogeneic; IV	1 × 10^6^; 1 dose	250	6	Yes	June 2020
NCT03186417	I	MPCs	2017	USA	Recruiting	During onset	Allogeneic; IV	2, 4 or 6 × 10^6^; 1 dose	20	24	Yes	December 2020
NCT03618784	I/II	UC	2018	Korea	Recruiting	Refractory	Allogeneic; IV	5.0 or 10 × 10^7^/patient; 3 doses	33	4	Yes	April 2021
NCT03691909	I/II	AD	2018	USA	Recruiting	Stable treatment	Autologous; IV	Unknown	15	12	No	June 2020
NCT03828344	I	UC	2020	USA	Active, not recruiting	Refractory	Allogeneic; IV	0.75 or 1.5 × 10^6^; 1 dose	16	12	Yes	September 2020
NCT04170426	I/IIa	AD	2020	USA	Active, not recruiting	DMARD-resistant	Autologous; IV	2.0 or 2.86 × 10^6^; 1 dose or 3 doses, every 3 days	54	12	Yes	December 2023

AD, adipose tissue; BM, bone marrow; UC, umbilical cord; MPCs, mesenchymal progenitor cells; DMARD, disease-modifying antirheumatic drug; IV, Intravenous; N/A; Not applicable, meaning trials without Food and Drug Administration-defined phases.

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
