# Peer review of "Mesenchymal Stem/Stromal Cells for Rheumatoid Arthritis Treatment: An Update on Clinical Applications"

_cells, 2020, doi:10.3390/cells9081852_

Round 1

Reviewer 1 Report

This review, ‘Mesenchymal Stem Cells in the Treatment of Rheumatoid Arthritis’ is an extensive, very insightful manuscript that includes the past, present, and future aspects of MSC treatment in RA. I have gone over several times the context and description of each trial, and the description is highly detailed. I have the following comment or recommendation. 

- In Table 2 and 3, the acronym UC represents umbilical cord whereas the main text it stands for umbilical cord blood; the former uses MSC driven by the umbilical cord tissue itself. The source could be different between trials; this should be clarified throughout the manuscript. 

For instance, NCT02221258 is umbilical cord blood-driven (NCT03618784  as well)

- On verification via variable sources, In Table 3, NCT03618784, 1 dose —> 3 doses. This may affect the description in the text on line 328, ‘Interestingly’, for NCT03618784 is not a repeated phase 1 trial of NCT02221258.

  • On verification, Line 318, ‘/kg of body weight’ would need to be removed.
  • On verification, Line 428, ‘although not recruiting yet’ —> ‘and recruiting’

Thank you for your contribution. 

Author Response

Comments and Suggestions for Authors

This review, ‘Mesenchymal Stem Cells in the Treatment of Rheumatoid Arthritis’ is an extensive, very insightful manuscript that includes the past, present, and future aspects of MSC treatment in RA. I have gone over several times the context and description of each trial, and the description is highly detailed. I have the following comment or recommendation. 

  • In Table 2 and 3, the acronym UC represents umbilical cord whereas the main text it stands for umbilical cord blood; the former uses MSC driven by the umbilical cord tissue itself. The source could be different between trials; this should be clarified throughout the manuscript. For instance, NCT02221258 is umbilical cord blood-driven (NCT03618784 as well)

We very much appreciate the Reviewer´s suggestion on this point and accordingly in the revised manuscript the acronym UC for umbilical cord has been used (Line 105). To clarify the specific source of the UC-derived MSCs either whole tissue or blood has been clearly stated within the text (Lines 202, 246, 270, 345,393, 409, 424, 447, 470, 488 and figure 1 and 2 legends).

  • On verification via variable sources, In Table 3, NCT03618784, 1 dose —> 3 doses. This may affect the description in the text on line 328, ‘Interestingly’, for NCT03618784 is not a repeated phase 1 trial of NCT02221258.

Reviewer 1 is correct. we have double-check and correct the number of doses in NCT03618784 clinical trial in the revised manuscript (Line 452) and in the accompanying tables and figures.

  • On verification, Line 318, ‘/kg of body weight’ would need to be removed.

Reviewer’s comment is correct. Accordingly ´/kg of body weight´ in NCT03618784, NCT01547091 and NCT02221258 clinical trials has been corrected in the revised manuscript (Lines 247, 346 and 452), tables and figures.  

  • On verification, Line 428, ‘although not recruiting yet’ —> ‘and recruiting’.

Reviewer’s comment is correct.  The recruiting status of the NCT02221258 trial in the revised manuscript has been updated.

Reviewer 2 Report

The review entitled: "MESENCHYMAL STEM CELLS IN THE TREATMENT OF RHEUMATOID ARTHRITIS." is an up to date resume of the state-of-the-art of a potential MSC therapy for rheumatoid arthritis. It also points to the necessity that further research is needed to improve its potential value. The literature list is not biased and includes most, if not all, relevant papers.

Author Response

We very much appreciate Reviewers 2 comments on the manuscript submitted.

Reviewer 3 Report

Lopez-Santalla M 
et al propose an interesting review on the therapeutic efficacy of mesenchymal stem cells for rheumatoid arthritis, analyzing the different outcomes of clinical trials that exist at today. Even that the thematic is of broad interest there are some issues that should be corrected before publication. 1.- The title is too general. It should change to something more specific such as “MSC for RA treatment: An update on clinical applications” or similar. 2.- The preclinical studies section should also include novel approach propose to improve MSC therapeutic efficacy for RA treatment. Indeed, hypoxia, proinflammatory cytokines and PPARb/g inhibition have been propose to improve MSC therapeutic efficacy. In particular Luz-Crawford et al (ARD. 2016) and Xian Xu et al (ARD, 2020) have interesting preclinical studies evaluating different strategies to improve MSC therapeutic efficacy for RA treatment. It should be interesting to discuss this part in the first section. What about extracellular vesicles? It should be interesting if the author could discuss about it also in this section Cosenza et al, Theranostics, 2018). Minor observations 3.-Resolution of figure 1 and 2 is not very good. Authors should change the colors.

Author Response

Comments and Suggestions for Authors

Lopez-Santalla M et al propose an interesting review on the therapeutic efficacy of mesenchymal stem cells for rheumatoid arthritis, analyzing the different outcomes of clinical trials that exist at today. Even that the thematic is of broad interest there are some issues that should be corrected before publication.

  • The title is too general. It should change to something more specific such as “MSC for RA treatment: An update on clinical applications” or similar.

We very much appreciate the Reviewer´s suggestion.  The title of the revised manuscript has now changed to ‘Mesenchymal stem/stromal cells for rheumatoid arthritis treatment: An update on clinical applications”.

  • The preclinical studies section should also include novel approach propose to improve MSC therapeutic efficacy for RA treatment. Indeed, hypoxia, proinflammatory cytokines and PPARb/g inhibition have been propose to improve MSC therapeutic efficacy. In particular Luz-Crawford et al (ARD. 2016) and Xian Xu et al (ARD, 2020) have interesting preclinical studies evaluating different strategies to improve MSC therapeutic efficacy for RA treatment. It should be interesting to discuss this part in the first section. What about extracellular vesicles? It should be interesting if the author could discuss about it also in this section Cosenza et al, Theranostics, 2018).

We have carefully reconsidered Reviewer´s suggestion on adding novel approaches to improve MSC therapeutic efficacy for RA treatment. In the revised manuscript, a comment on MSC-derived EVs and the used genetically engineered MSCs in preclinical studies of RA have been included (Line 141). As suggested by the reviewer, we have also included the new ChiCTR-ONC-16008770 clinical trial published recently by Xian Xu et al in the new version of the manuscript (line 293). We have also updated the table II and the Figure 1 accordingly.

  • Minor observations

Resolution of figure 1 and 2 is not very good. Authors should change the colors.

As suggested by the Reviewer, figures 1 and 2 are now in greyscale TIFF format in the new version of the manuscript. If necessary we can further edit the figures to improve their quality.

Round 2

Reviewer 3 Report

Authors have raised all my concerns. I recommend to perform an editing of English.